# Semantic Routing via Autoregressive Modeling

**Eric Zhao**
UC Berkeley and Google Research
eric.zh@berkeley.edu

**Pranjal Awasthi**
Google Research
pranjalawasthi@google.com

**Zhengdao Chen**
Google Research
zhengdaoc@google.com

**Sreenivas Gollapudi**
Google Research
sgollapu@google.com

**Daniel Delling**
Google
delling@google.com

## Abstract

We study learning-based approaches to semantic route planning, which concerns producing routes in response to rich queries that specify various criteria and preferences. Semantic routing is already widely found in industry applications, especially navigational services like Google Maps; however, existing implementations only support limited route criteria and narrow query sets as they rely on repurposing classical route optimization algorithms. We argue for a learning-based approach to semantic routing as a more scalable and general alternative. To foster interest in this important application of graph learning, we are releasing a large-scale publicly-licensed benchmark for semantic routing consisting of real-world multi-objective navigation problems—expressed via natural language queries—on the richly annotated road networks of US cities. In addition to being intractable with existing approaches to semantic routing, our benchmark poses a significant scaling challenge for graph learning methods. As a proof-of-concept, we show that—at scale—even a standard transformer network is a powerful semantic routing system and achieves non-trivial performance on our benchmark. In the process, we demonstrate a simple solution to the challenge of scaling up graph learning: an autoregressive approach that decomposes semantic routing into smaller "next-edge" prediction problems.

## 1 Introduction

The route planning problem arises in diverse applications including navigating transportation networks and supply chain logistics (Dantzig and Ramser, 1959). Route planning is classically formulated in the language of graph theory as searching for paths between a source and a destination that optimize a single static objective (Dijkstra, 1959), usually travel time. In practice, route planning systems are not as simple. For example, navigation services like Google Maps do not only optimize for travel time—they allow drivers to specify multiple modes of transportation, add intermediate stops, and convey preferences about avoiding roads with bike lanes (Ratliff et al., 2009). Modern routing systems are better described as performing *semantic routing*: a contextual and multi-objective generalization of route planning that involves processing a rich user query that specifies not only a source and destination but also various criteria—such as bike lanes preferences—and responding with a satisfactory route (Delling et al., 2017).

Current routing systems implement semantic routing capabilities in an ad hoc way, customizing classical route optimization algorithms to support specific and limited types of route criteria. For example, the technique of "cost modifiers" rescales edge weights according to whether the edge type is preferred/dispreferred, such as doubling travel time along small roads when it is indicated that small roads should be avoided (Ratliff et al., 2009), and then runs a standard shortest-paths

38th Conference on Neural Information Processing Systems (NeurIPS 2024).

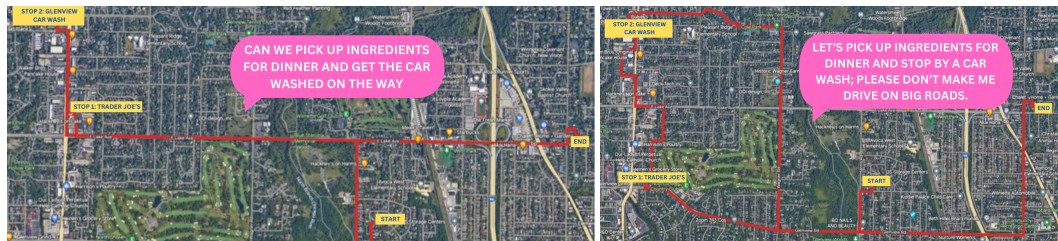

Figure 1: Examples of semantic routing tasks from our benchmark that involve navigating between two locations in the Glenview neighborhood of Chicago. The user query is specified in pink, while we outline the path of the optimal route (according to our automated evaluation scheme) in red.

algorithm. As industry routing services transition to offering increasingly rich experiences, it has become clear that the current paradigm is unscalable. First, this ad hoc approach can only support the limited types of route criteria that can be efficiently achieved through classical graph algorithms; for example, cost modifiers only support "linear" criteria (Ratliff et al., 2009). Second, modern routing systems are complex ensembles of combinatorial algorithms and expensive to maintain; supporting new features requires adding new customized algorithms into an already messy ensemble and even then different route criteria can often not be efficiently combined. Third, current systems cannot take full advantage of rich graph metadata as they require explicit instructions on what and how to use each data field (Cormen et al., 2022); this is a major opportunity cost as, for example, modern road network metadata include live road conditions, the businesses along each road (OpenStreetMap contributors, 2017), and even the locations and types of fruit trees (Welty and Phillips, 2013). One can directly observe how these limitations manifest in industry services. For example, Google Maps allows for specifying an intermediate stop if an explicit address is given but cannot automatically plan a short stop to purchase dental floss; it can provide routes that use public transportation but not when an intermediate stop is specified. In this paper, we study the open question of building a semantic routing system that is truly scalable and offers general open-ended capabilities.

We argue for a learning-based approach to semantic routing, where systems are trained on historical, and potentially simulated, data consisting of tuples of queries, rich graphs, and satisfactory routes. A learning-based approach avoids all three key limitations of the current paradigm. It transfers the complexity of semantic routing from handcrafting ensembles of combinatorial graph algorithms to instead creating a rich dataset that demonstrates desired capabilities. Moreover, as we will demonstrate, learned models can approximately fulfill route criteria that may not admit efficient classical solutions and take advantage of available rich graph metadata that may be impractical to manually engineer support for. Semantic routing also exhibits the usual hallmarks of a practical machine learning application: though undesirable, having latency and errors in a semantic routing system is not typically catastrophic or dangerous, unlike higher-stakes applications such as autonomous driving. With the cost of deploying large neural networks plummeting and the toolkit of graph learning methods exploding in recent years, we believe that learned semantic routing systems are now a viable short-term goal. Our contributions are two-fold and aimed at fostering interest in this important application of graph learning.

1. First, we release a large-scale publicly-licensed benchmark for semantic routing, which can be found at github.com/google-research/google-research/tree/master/semantic_routing. This benchmark, and the remainder of our paper, focuses on real-world multi-objective navigation problems that involve providing route suggestions to drivers. These problems include providing efficient routes for errand running, planning recreational day trips, avoiding highway driving, and providing waypoint routes that make stops by, e.g., grocery stores. Each problem in this benchmark is a tuple consisting of (1) a subgraph of a real-world road network with rich metadata including local businesses, sourced from the OpenStreetMap project (OpenStreetMap contributors, 2017), and (2) an unstructured natural language user query. Our benchmark also includes an automated evaluation mechanism that, given a problem tuple, scores candidate routes. Consisting of 1,000,000+ user queries and 300,000+ miles of real-world road networks, our benchmark provides a comprehensive platform for evaluating semantic routing systems and tests capabilities that are intractable with existing methods.

2. Second, we demonstrate the viability of learning-based semantic routing systems with a proof-of-concept. Specifically, we show that—when done at sufficient scale—training even a standard

| |
|---|
| Find a suitable place for dog walking and a post box along the way. |
| {need: (dog park or park) and (post box)} |
| Help me plan a kid-friendly adventure through town, with all places no more than a 20-minute drive away. |
| {visit: (game store or video game store or toy store or arcade)} |
| Please plan a route for me that includes a stop at a bakery and a car repair shop, avoiding highways whenever possible. |
| {visit: (bakery) and (car repair shop), dislikes: highways } |

Figure 2: Examples of queries drawn from our benchmark, demonstrating an errand route planning task (top), a trip planning task (middle), and the combination of a waypoint routing task and highway avoidance task (bottom). Annotated labels are underneath queries in gray and are hidden to models.

transformer network on our benchmark's dataset creates a powerful semantic routing system, with a 60-70% success rate on benchmark tasks even on previously unseen user queries and in a previously unseen American city. As existing semantic routing methods entirely fail on our benchmark—even when implemented with unfair advantages—our proof-of-concept also establishes the first non-trivial baseline of our benchmark. A major challenge we encountered was scaling up graph learning methods to large road graphs (e.g., 10,000 vertices and 25,000 edges); we identified an *autoregressive approach* to be critical: performing repeated next-edge predictions to unroll a route instead of predicting an entire route at once, allowing us to only provide the model a small local subsection of the road network at a time. This approach is similar to that of a language model, except instead of unrolling a sentence by predicting tokens, we unroll a route by predicting edges.

## 1.1 Related Work

**Route Planning in Navigation.** Traditional routing tasks are approached as finding shortest paths within road networks and can be solved with dynamic programming algorithms such as Dijkstra's method (Dijkstra, 2022) and its generalizations (Shimbel, 1954; Bellman, 1958; Ford, 1956). Their efficiency has been improved via priority-queue-based implementations (Goldberg, 2001; Meyer, 2001), bidirectional search (Sint and de Champeaux, 1977), A$^*$ search (Hart et al., 1968), and contraction hierarchies (Bast et al., 2016; Delling et al., 2009; Bauer et al., 2010) that specifically make use of the near planar structure of the road networks. However, incorporating custom contextual information into these algorithms is challenging, requiring solving a (often NP-complete) combinatorial problem per context. For a very narrow set of contextual routing tasks where the "cost" of a route can be decomposed into sum of its component edges, it is possible to define context-dependent *cost modifiers* on edges such that the running shortest-paths on the new costs results in satisfactory solutions (Ratliff et al., 2009). However, this approach is not possible for most routing tasks of interest and—even when available—still requires one to solve the challenging problem of learning how to design good cost modifiers from complex natural language queries. A more generally available alternative is to run unguided search algorithms such as A$^*$ search or spectral methods based on electrical flows (Sinop et al., 2021) to compute a large set of candidate routes from source to destination, and then extract the best candidate. However, this approach needs one to have a good way of scoring candidate routes and may require iterating through intractably many candidates to find a solution.

**Learning on Graphs.** In recent years, many data-driven approaches have been explored for combinatorial problems on graphs. The works of Xu et al. (2019, 2020) show that graph neural networks (GNNs) can learn dynamic-programming-based algorithms; Veličković et al. (2019); Ibarz et al. (2022) create GNN based neural learners to simulate various combinatorial algorithms by "hinting" them with intermediate computations of the algorithms. Other works have explored the custom models, such as pointer networks and attention-based GNNs, for solving graph problems such as the traveling salesman (Kool et al., 2018; Vinyals et al., 2015; Khalil et al., 2017; Nowak et al., 2017; Deudon et al., 2018). However, these approaches are all empirically tested only at small scales: graphs around 100 nodes large. In comparison, the semantic routing applications we explore in this paper concern road networks that are—at smallest—10,000 vertices large; naively scaling graph learning algorithms to this setting would require an impractically expensive distributed GNN system. One exception to the small scale studies in previous works is Graves et al. (2016), who conducted routing experiments on the London underground system using a highly specialized and complex variant of a

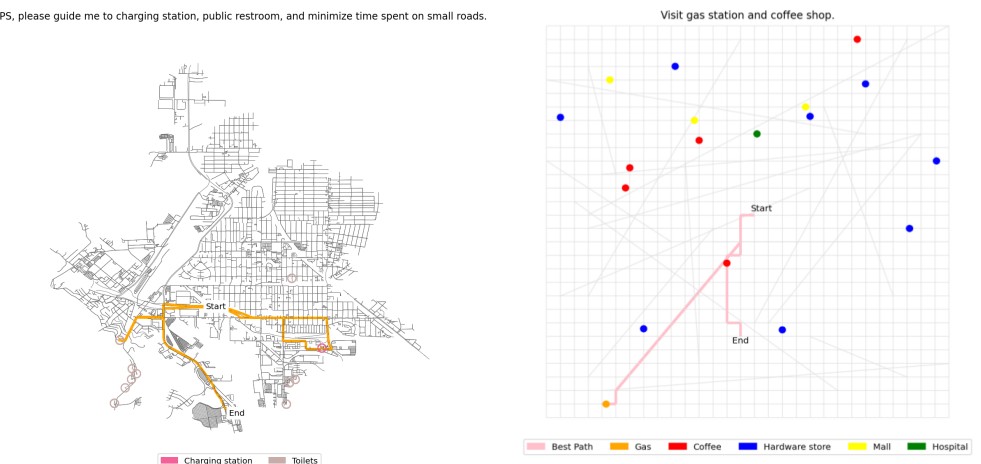

Figure 3: Example of semantic routing tasks from our main benchmark (left) and our grid-world (right), visualized in false-color. The optimal route is illustrated in orange/pink and colored dots denote points-of-interest.

differential neural computer. In contrast, we present in Section 3 a simple approach to scaling graph learning methods to large-scale semantic routing problems: train on an autoregressive objective so that the semantic routing problem decomposes into smaller next-edge prediction problems where it suffices to only consider a local neighborhood of the graph.

**Autoregressive Models.** Autoregressive approaches based on the transformer model (Vaswani et al., 2017) have achieved remarkable progress for generative tasks from language modeling to image and video generation. The ability of such models to generate sequences via the next-token-prediction paradigm and to incorporate multi-modal information makes them a natural fit to tackle semantic routing. Moreover, when trained at scale and augmented with prompting techniques or external tools and operations, autoregressive models appear to demonstrate capabilities of planning (e.g., Wei et al. (2022); Huang et al. (2022); Schick et al. (2023); Yao et al. (2023); Valmeekam et al. (2023)). We take inspiration from autoregressive methods to approach semantic routing, re-discovering strong parallels between the domains of routing and language.

## 2 A Benchmark for Semantic Routing

We are releasing a public dataset of semantic routing problems concerning complex navigation tasks on American roads. To facilitate discussion, we will formalize a semantic routing problem as consisting of a graph $G = (V, E)$ with vertices $V$ and edges $E$, a source and destination pair $u, v \in E$, and a query $s \in \Sigma^*$ where $\Sigma^*$ denotes all strings of alphabet $\Sigma$. We can denote the metadata associated with each edge in $G$ using the map $\theta : E \to \Xi$, where $\Xi$ is the space of possible metadata. Note that a single query $s$ may communicate multiple subtasks and objectives. The goal of a semantic routing problem is to find a path $p \in E^*$ from source to destination that "satisfies" the query $s$. We can formalize this goal as finding a path $p$ that starts at the source $u$, ends at the destination $v$, and maximizes $f(s, \{\theta(e)\}_{e \in p})$ where $f : \Sigma^* \times \Xi^* \to \mathbb{R}$ is a scoring function and $\{\theta(e)\}_{e \in p}$ is the metadata associated with each edge in the path. In our benchmark, each query $s$ is a natural language user query, so $\Sigma$ can be understood as the English alphabet. Our graph $G$ is a road network where each edge corresponds to a road segment, and edge metadata $\theta$ contains road information such as colocated businesses and the speed limit.

### 2.1 Benchmark Overview

This benchmark evaluates semantic routing algorithms on whether they can successfully return a satisfactory route when provided a natural language semantic routing query and a road graph on which to navigate. To this end, we collected and labeled a large dataset of natural language semantic routing queries and curated open-source data from OpenStreetMap contributors (2017) on the road

| Method Name | Waypoint Routing Success Rate |
|---|---|
| Cost modifiers | $0\% \pm 0\%$ |
| Electrical flows (4096 routes) | $1.3\% \pm 1.3\%$ |

Table 1: This table depicts the success rate of classical semantic routing methods (named in the "Baseline" column) on waypoint routing task sampled from our benchmark. We define a method as being successful if it returns at least one route that visits points-of-interest that together fulfill each part of the user's query. The "cost modifiers" method returns every route that can be elicited by a cost modifying rule (Delling et al., 2009); the "Electrical flows" method returns 4,096 routes sampled with the electrical flows algorithm (Sinop et al., 2021). $\pm$ denotes standard error ($n$=100).

networks and points-of-interest (facilities, businesses, restaurants, etc.) of major urban centers in the United States. The benchmark is released under an Apache 2.0 license; see supplemental materials. Structured similarly to a reinforcement learning benchmark, it is accessible as a software package with three main functions: (1) generates semantic routing tasks by randomly drawing a user query from a data bank and choosing the road graph of an American city, (2) given a semantic routing task, featurizes the task by generating feature arrays that describe the road graph in the form of a Tensorflow record (Abadi et al., 2015), and (3) given a semantic routing task and a candidate solution, returns a numerical score that describes estimated route quality. Examples of these functionality are provided in Python notebooks available in the previous link.

The benchmark consists of over 1 million natural language queries and navigation tasks drawn from an over 11 million vertex graph representing 375,000 miles of real-world road networks and 1.3 million points-of-interest in 23 major US cities. Figure 1 illustrates samples of semantic routing tasks from our benchmark and their solutions.

**Benchmarked Tasks.** Our benchmark measures semantic routing capability with important real-world problems that involve navigating drivers on roads. The majority of tasks in our benchmark involve a variant of waypoint routing: routing tasks that require planning out (usually multiple) intermediate stops on one's route. For example, a user might ask to get coffee and drop off a package en route to work. Two types of waypoint routing tasks appear in our benchmark: where the user specifies a specific location category to visit (e.g., a grocery store), or where the user specifies a need or errand that they need to complete en route (e.g., buy a water bottle). The latter allows for more flexibility as, for example, a water bottle can be purchased at many types of locations including both grocery stores and gas stations. Fulfilling a waypoint query involves up to four expensive steps:

1. Identifying points-of-interest that fulfill the request (e.g., nearby coffee shops and mail boxes).
2. Choosing which points-of-interest to stop by (e.g., picking a coffee shop and mailbox to visit).
3. Picking an order to make the stops (e.g., determining that getting coffee first would be faster).
4. Efficiently navigating through stops by minimizing travel time or fulfilling additional objectives.

One fifth of the tasks in our benchmark require also taking into account a driver's personal comfort preferences, i.e., whether a driver has an aversion to highways or a preference against driving on small local roads. Another subset of tasks in our benchmark involve planning day trips, i.e. itinerary planning. In these tasks, the user query specifies a time budget and the type of day trip that the user wants to go on; the goal is to return a route that takes the user to as many relevant points-of-interest as possible within the time budget. Whereas waypoint routing is a variant of the traveling salesman and set cover problems, trip planning is a variant of the knapsack problem.

We provide examples of each of these tasks in Table 2. A challenging aspect of our benchmark is that a single problem typically involves multiple tasks; for example, a single query might describe multiple errands that need to be run en route and also an aversion to highways. On average, a single query conveys 1.8 distinct tasks or objectives, with a maximum of up to 5 tasks per query.

**Baselines.** In this section, we adapt classical semantic routing methods to our benchmark and evaluate them on their success rate on waypoint routing task. Here, we define a route as being successful if it visits points-of-interest that together fulfill every request in the user's query; we will ignore further considerations like travel time for simplicity. As noted in Section 1, we expect the

inherent limitations of classical methods to result in poor performance and will therefore generously adapt these methods to afford them unfair advantages.

First, we consider the method of *cost modifiers* (Delling et al., 2009) where—depending on the user query—we rescale travel times along each edge according to the edge's metadata, and then apply a classical shortest-paths algorithm to find a route. To upper bound the performance of systems that use cost modifiers on our benchmark, for every semantic routing problem, we perform a brute-force search over the routes returned by each possible cost modifier scheme (i.e., edge rescaling rule) and check if any of the routes is successful; if so, we mark down the overall method as being successful. Second, we consider a guess-and-check method that uses *electrical flows* (Sinop et al., 2021) to generate a large and diverse set of candidate routes and check if any of the routes is successful; once again, we mark down the method as being successful if such a route is found.

The performance of these methods, as shown in Table 1, is unsurprisingly poor. Whereas waypoint routing problems are provably unsolvable with cost modifier methods, the electrical flows method is similarly doomed due to the intractably large search space of possible routes from source to destination. For a baseline level of performance on our benchmark, we refer readers to Section 3.

## 2.2 Technical Details

**Road network data.**  We curated and sourced road network data and points-of-interest data from the OpenStreetMap project for the American cities of New York City, Santa Barbara, Miami, Cambridge (MA), Providence, San Francisco, Orlando, Boulder, Albuquerque, Portland, Madison, New Orleans, Chicago, Palo Alto, Philadelphia, Austin, San Jose, Memphis, Denver, Washington DC, Seattle, Sacramento, Santa Cruz, and Berkeley. Our curated road networks embed rich metadata in every road segment, including each segment's road type (e.g., interstate, service road), number of lanes, speed limit, length, estimated average transit time, geometry (start and end coordinates), and a list of points-of-interest that can found along the road segment. Every point-of-interest in our road networks exists in the real world and is annotated with its real-world name and assigned to one of over two hundred point-of-interest categories, such as "gas station" or "mailbox".

Our benchmark produces semantic routing problems by randomly selecting pairs of source and destination edges within the road network of Berkeley, California; for tasks involving trip planning, the destination is the same as the source. We perform rejection sampling to omit source-destination pairs if the optimal route between the two requires traversing more than 600 road segments. Due to its diverse composition, we chose the city of Berkeley as our benchmark's default hold-out city, meaning that the training datasets that accompany our benchmark do not contain any examples of semantic routing in Berkeley. This means that our benchmark evaluates semantic routing systems on a road network they have never encountered and with points-of-interest and roads they have never seen in training. To simulate live traffic conditions on our road networks, we randomly rescale each road's default transit time in every new semantic routing problem instance.

**Query dataset.**  We collected one million natural language user queries through a combination of human labeling and language model augmentation. The first step in our process produced 1,000+ distinct types of semantic routing queries by manually taxonomizing common road situations (e.g. ran out of gas, passenger spilled drink in backseat), and writing representative queries for each situation in the taxonomy. An industry language model (OpenAI, 2023) was then applied to augment these queries into 10,000+ natural language variations. Multi-objective queries were then formed by randomly merging together combinations of up to 5 queries, where we again applied a language model to refine the writing of combined queries. The resulting queries were spot-checked for correctness and are depicted in Figure 2 and Appendix D. Tour planning queries were not merged with other query types.

To define an evaluation rule for our benchmark, i.e. the route scoring function $f$, we label every query in our dataset with a structured interpretation of its content. For example, queries that specify an aversion to driving on highways have a special flag in their label indicating such. For every waypoint routing task communicated in a query, the query's label contains a list of point-of-interest categories that would satisfy the task; for example, a "get waterbottle" task might be labeled with the list ("grocery store", "gas station", etc.). Similarly, for every trip planning task communicated in a query, the query's label contains a list of point-of-interest categories that the driver might want to visit on the requested trip. Queries are human labeled, though when two queries are rewordings

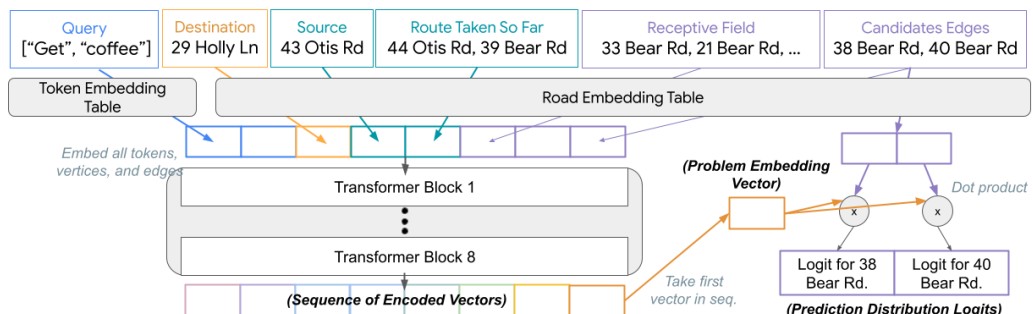

Figure 4: Architecture of proof-of-concept semantic routing model: a transformer network trained on an autoregressive objective. Arrows illustrate the flow of data from input to next-edge prediction.

of one another, we may only label one query and propagate its label to the other. Figure 2 depicts random examples of queries and their labels.

**Automated route evaluation.**    Although it is possible to use human raters to evaluate routing systems on our benchmark's collection of semantic routing problems, we also provide an automated system for evaluating candidate routes against problems in our benchmark. Our system—which simulates a generic user's preferences with standard heuristics—returns various metrics about the quality of the route, including an overall numerical score and task specific performance metrics like the waypoint routing success metric of Table 1. Our evaluation system's overall score metric should be understood as providing a ground-truth ordering over routes in terms of goodness; similarly, the ground-truth optimal route is the route maximizing this score. We define the overall score as follows. First, we refer to the problem's query label and identify all waypoint requests and points-of-interest that would satisfy the requests. If the route does not visit points-of-interest that satisfy all waypoint requests, an infinitely large penalty is subtracted from the score. The total seconds of travel time is also deducted from the overall score; although if the user specifies an aversion/preference for driving on a particular road type, travel time on such roads is increased/decreased by a factor 5/0.5. In trip planning tasks, a reward of 1,000 is added to the score for every relevant and unique type of point-of-interest visited. When a time budget is communicated in the query, an infinitely large penalty is subtracted from the score if the time budget is exceeded by more than a fourth.

Although this evaluation scheme can be succinctly stated, its optimization is non-trivial and generalizes various combinations of traveling salesman, shortest path, set cover, and knapsack problems. Moreover, this reward scheme leads to empirically high-quality results as demonstrated in Figure 1.

**Training dataset.**    We are also releasing a training dataset consisting of 3 million pairs of semantic routing problems generated from our benchmark and their ground-truth optimal routes. We brute-force solved for these routes by peeking at the query labels and applying an ensemble of combinatorial algorithms, mimicking a very inefficient classical semantic routing system. Beyond query labeling, this requires 0.3 CPU hours per route.

**Grid-world test bed.**    To provide a more flexible test bed for semantic routing algorithms, we are also releasing a purely synthetic semantic routing benchmark. This benchmark simulates a large grid graph with highway roads and populates it with artificial points of interest, as illustrated in Figure 3. Each grid world generated is of size 30x30 and consists of 3560 road segments; these graphs can be understood as randomly sampled subgraphs of an infinitely large grid world, taken around a navigation task's source vertex. The query dataset for this grid world consists of simple template-generated natural language queries and represents only waypoint routing tasks. Some experiments in Section 3 are performed on this grid-world benchmark.

## 3    A Proof-of-Concept Autoregressive Model

One challenge that has limited the adoption of graph learning methods in routing applications is that routing is a naturally large-scale decision-making process. Road networks form highly connected

| | |
|---|---|
| Success Rate (Waypoint Routing, Errands) | $92.0\% \pm 3.7\%$ |
| Success Rate (Waypoint Routing, Locations) | $38.0\% \pm 6.6\%$ |
| Success Rate (Waypoint Routing, Mixed) | $34.0\% \pm 6.9\%$ |
| Success Rate (Trip Planning) | $68\% \pm 4\%$ |
| Median Excess Penalty (Waypoint Routing, Mixed) | 0.8  seconds |
| Median Total Travel Time (Waypoint Routing, Mixed) | 509  seconds |

Table 2: The task-level performance metrics of the proof-of-concept autoregressive transformer model on our benchmark. The evaluated problems represent various semantic routing tasks with queries drawn from a holdout test set and conducted on a road network from a previously unseen city (Berkeley, CA). "Waypoint Routing Success Rate" is the fraction of waypoint routing problems that are successfully completed (see Table 1). "Errands" are waypointing needs that can be fulfilled by multiple point-of-interest categories (e.g., buy water bottle), "locations" are those that correspond to a specific point-of-interest category (e.g., gas station), and "mixed" refers to waypoint routing problems consisting of both errand and location subtasks. "Trip Planning Success Rate" is the fraction of trip planning tasks that are completed under the time budget and where the predicted route provides at least half the experiences of an optimal itinerary. "Excess Penalty" is the median gap between the achieved overall route score and best possible score, conditioned on successfully completed waypoint routing problems; this penalty can be understood as total travel time (in seconds) rescaled by 5x when transiting along an unfavorable road type. Standard error is denoted by $\pm$.

graphs, and even a 10 kilometer stretch of road involves, on average, 600 different road segments (OpenStreetMap contributors, 2017). Meanwhile, popular deep learning methods for graph-structured data such as GNNs are often difficult to scale, leading to an impression that graph learning methods are an impractical solution to semantic routing.

In this section we demonstrate a simple graph learning method that serves as a strong baseline for semantic routing tasks. Our key observation is that we can train autoregressive networks on a natural next-vertex prediction objective to *unroll* routes. We show that, by decomposing semantic routing into next-vertex subproblems, it is possible to construct high-quality routes while only needing to engage a small neighborhood of the road network at each point in time. This approach allows standard transformer networks to achieve strong results on our semantic routing benchmark.

## 3.1   Autoregressive Modeling

Autoregressive learning is an approach to learning generative models where one trains a network on the task of predicting the next element of a string (Brown et al., 2020), which can be a sentence in language generation or, in semantic routing, an edge. These networks can be used to generate strings by unrolling a solution path through repeated next-*element* predictions or, in the case of semantic routing, through repeated next-*edge* predictions. From a probabilistic standpoint, this approach circumvents the need to estimate the distribution of the entire solution path by decomposing it into a sequence of conditional distributions via the chain rule.

Despite their greedy nature, autoregressive predictors have demonstrated remarkable success in language generation and showcased their ability to effectively navigate complex decision spaces (Huang et al., 2022; Bubeck et al., 2023). Moreover, they are proven theoretically to be universal learners if trained on adequate data (Malach, 2023). These results suggest autoregressive models to be a viable graph learning solution to semantic routing: one can avoid needing to scale graph learning algorithms to large road networks if they can decompose routing into next-edge prediction problems where only a small neighborhood of the graph is relevant at any single time.

## 3.2   A Proof of Concept

We trained a standard transformer network on our benchmark's training dataset. By truncating the example routes in the dataset at random positions, we optimized our model on an autoregressive objective of predicting subsequent edges. As inputting entire road networks into our model was impractical due to its $O(|V|^2)$ memory footprint, we instead input a small neighborhood of the road network around the head of each truncated route. The resulting model fits on a single GPU.

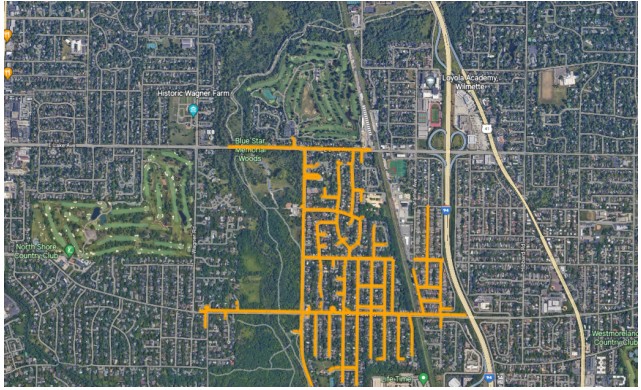

Figure 5: Visualization of a receptive field consisting of 512 road segments; a receptive field is a neighborhood that is the taken around the head of a partial route. As discussed in Section 3.3, our semantic routing model observes a 512-edge receptive field instead of the entire road network.

We report the performance metrics of this model in Table 2. The model successfully interprets and fulfills roughly 4 in 5 waypoint routing tasks; recall that this evaluation is performed on previously unencountered queries and a previously unseen city's road network. The model incurs a median excess travel time of less than 0.2%; this is including a 5x travel-time penalization for unnecessary travel on roads (e.g., highways) that the user query specifies an aversion against. Our model also provides a satisfactory itinerary for 2 in 3 trip planning queries. Although it is trivial to compare these results to those of classical methods, we refer readers to Appendix D for random examples of our model's predictions for an intuitive sense of these performance metrics. We believe these figures to be a strong demonstration of the viability of learning-based semantic routing; these success rates on challenging combinatorial problems expressed in natural language is non-trivial in any application domain. Moreover, we are confident that our proof-of-concept is far from an upper bound of what is achievable with learned semantic routing models.

### 3.3 Technical Details

**Model description.**    Our proof of concept implements a simple decoder-only transformer network (Vaswani et al., 2017; Brown et al., 2020) which we train using a standard cross entropy loss on the task of predicting the subsequent edge to a truncated route. Our network—as illustrated in Figure 4—takes as input a sequence of "tokens" that contain the user's query, the destination, the partial route that we are trying to predict the subsequent edge for, a set of possible subsequent edges, and the small neighborhood (the "receptive field") around the head of the partial route. To featurize the tokens in this sequence, we create a feature vector for each edge from their metadata, and process it with a multi-layer perceptron. This is in contrast to language models which learn an embedding table for their alphabet, which in our setting would correspond to learning an embedding vector for every segment in our road networks; we deviate from this embedding approach so we can take advantage of road segment metadata and encourage the model to generalize to edges not seen during training. Because of this, our proof-of-concept can provide high-quality routes in a city it has never encountered during training.

**Receptive field.**    Our autoregressive approach allows us to treat each next-edge inference as a local problem: instead of passing in the entire road graph into our model, we only pass in a small neighborhood around the head of the partial route. We refer to this small neighborhood as the receptive field of our model. Our proof-of-concept uses a 500-edge receptive field, which we illustrate the size of in Figure 5. In comparison, our navigation tasks are defined on 10,000-node (roughly 25,000 edge) subgraphs of urban road networks, meaning that our use of a receptive field reduced the memory footprint of our model by a factor of over 2,000.

**Inference strategy.**    Given a next-vertex prediction model, there are several approaches to obtaining a candidate path for a given routing task. The first approach is greedy decoding, a popular and computationally efficient baseline approach in language modeling (Chowdhery et al., 2022). To see

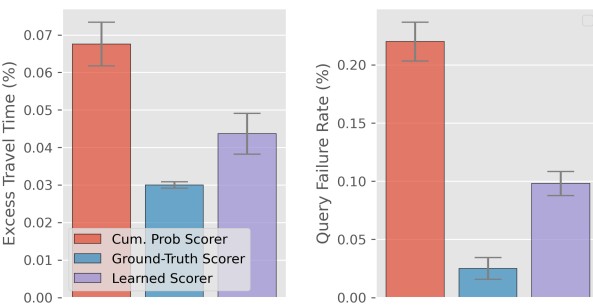

Figure 6: The semantic routing performance metrics of an autoregressive transformer under a 10-width beam-search algorithm. The route prediction is extracted from the final beam either using the model's logits (red), a secondary model (purple), or omnipotently choosing the best route (blue).

this, note that the time complexity of our next-vertex prediction framework is independent of the size of the road network. Rather, since each step requires quadratic computation in the length of the route so far, the full time complexity is cubic in the path length $O(|\mathrm{Route}|^3)$ for greedy decoding, with improved complexity possible through more efficient attention mechanisms (e.g. Wang et al. 2020; Zaheer et al. 2020; Dao et al. 2022).

Inspired by classical AI and natural language generation, we also apply *beam search* (Reddy, 1977) to our next-vertex prediction model during inference. Beam search returns a set of candidate paths (as many as the beam width) generated by the model and allows one to cleanly trade-off accuracy for inference complexity, with a run time of $O(\mathrm{BeamWidth}^2 \cdot |\mathrm{Route}|^3)$. We train a secondary *scoring model* to choose a final path from this candidate set. The scoring model is trained in a contrastive fashion, where ground-truth paths are taken from our training set as positive examples and random paths are sampled from the outputs of our model as negative examples. After embedding these routes with our next-vertex model, we train the scoring model as a 2-layer MLP with a logistic loss. At inference time, we select from candidate paths by choosing the one receiving the largest logits from the scoring model. In Figure 6, we ablate this scoring model and notice a significant drop in semantic routing performance. The main results presented in this paper, i.e. Table 2, use beam-search with a secondary scoring model.

### 3.4 Ablation Studies and Benchmark Demonstrations

Additional experiments, including ablation studies, can be found in Appendix A and illustrate how increasing receptive field sizes only marginally impacts model performance and evaluate how semantic route planning performance scales with model size and aspects of the road graph's topology. Appendix D lists randomly sampled examples of semantic route planning tasks from our benchmark along with the optimal route according to our benchmark's labeling system.

## 4 Discussion

We study a learning-based approach to semantic routing. To facilitate further research on this important graph learning application, we release a large-scale benchmark of real-world multi-objective navigation problems. We also introduce a proof-of-concept model to establish a competitive baseline for the benchmark and to demonstrate the viability of learning-based approaches. To this end, we formulated an autoregressive architecture that reduces semantic routing to the scalable problem of next-edge prediction. We refer readers to Appendix B for a continued discussion of open questions.

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

## A  Additional Experiments

In this section, we describe some additional brief experiments into the behavior of autoregressive transformers that are trained on semantic routing tasks.

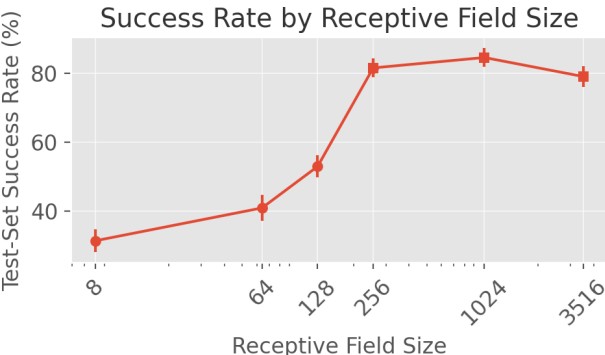

Figure 7: The grid world semantic routing performance of transformer networks of varying receptive field sizes. The largest possible receptive field size is the entire road network, which is 3516 edges.

**Receptive field experiment.**  In this experiment, we measure how model performance scales with receptive field size. As demonstrated by the strong performance of the proof-of-concept model, the greedy approach using receptive fields to make next-edge predictions with only local information is surprisingly effective. To validate that the use of a receptive field does not significantly degrade model performance, we consider train autoregressive transformer models on our grid-world benchmark, varying the size of each model's receptive fields. As depicted in Figure 7, there are no statistically significant gains in performance from passing in the entire road network (3560 edges) or even a large subset of it (1024 edges); rather, the gains from using a larger receptive field almost entirely disappear even when only a small fraction of the graph is included (256 edges).

We attribute this to two factors.  First, real-world road networks are intentionally designed to be amenable to human drivers, which, like our model, are often learning agents making greedy navigation decisions with only local information; similarly, businesses and facilities can usually be found in predictable places. Second, autoregressive methods have demonstrated significant utility even in applications where greedy approaches appear to be a poor fit. For example, recent findings increasingly indicate that language models—at scale—are capable of long-term planning despite their greedy nature (Huang et al., 2022; Valmeekam et al., 2023).

Figure 8: The grid world performance of an autoregressive transformer under beam-search with varying width.

**Beam search width tuning.** This experiment studies how beam search width affects the performance of autoregressive semantic routing models. To this end, we implement and train our autoregressive Transformer architecture on our grid-world semantic routing dataset, and evaluate the performance of the beam search inference strategy for varying beam widths. The results, depicted in Figure 8, demonstrate monotone but diminishing returns from increasing beam width.

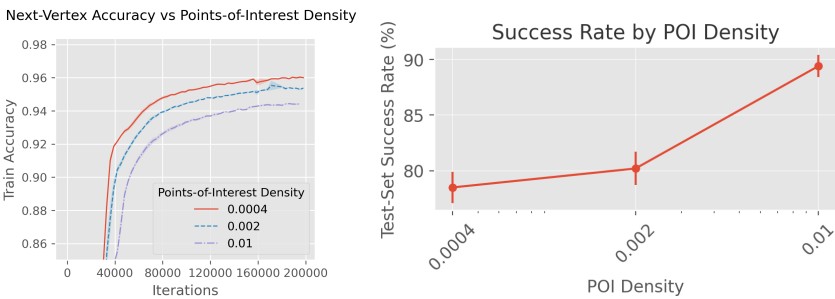

Figure 9: The next-edge prediction accuracy (left) and waypoint routing success (right) of autoregressive transformers on tasks in our grid-world benchmark.

**Point-of-interest density experiment.** In this experiment, we attempt to understand how the difficulty of fulfilling a query affects the difficulty of semantic routing and, in particular, waypoint routing tasks. One should expect that, when points-of-interest are extremely plentiful, waypoint routing reduces to finding the shortest path since the shortest path likely encounters the desired points-of-interest by chance. In this experiment, we train autoregressive transformers models on multiple variants of the grid-world benchmark where we vary the density of points-of-interest on our road network. The results plotted in Figure 9, indicate that while—as expected—a higher point-of-interest density results in higher waypoint routing success rates. However, we surprisingly also observe that a higher point-of-interest density makes next-edge prediction more difficult. Our interpretation of this apparent contradiction is that point-of-interest density indeed makes semantic routing easier, but also increases the number of reasonable routes that may exist; the resulting higher entropy distribution over solution routes results in a lower next-edge prediction accuracy. This highlights a trend we frequently observed while training autoregressive semantic routing models: relative differences in next-edge accuracy often significantly disagree with relative differences in system performance, such as waypoint routing success.

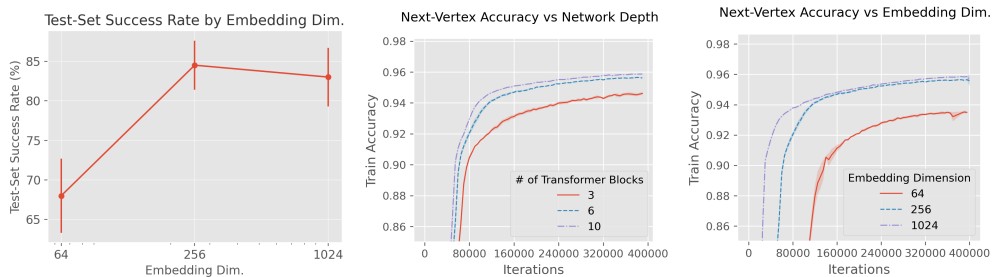

Figure 10: The semantic routing performance (left) and next-edge prediction accuracy (middle, right) of autoregressive transformers on our grid-world benchmark. On the left, both embedding dimension and network depth are rescaled. In the middle, only depth is rescaled. On the right, only embedding dimension is rescaled.

**A scaling study.** This experiment studies how network scale interacts with the performance of autoregressive semantic routing models. To this end, we implement our autoregressive Transformer architecture and train multiple copies of it on our grid-world semantic routing dataset, where each copy is configured with a different model capacity. We rescale our model in two directions. We first vary the depth of our network from its default depth of 6 transformer blocks to either 3 or 10 blocks deep. We separately vary the width of our network from its default embedding dimension of 256 to

either a dimension of 64 and 1024; we follow the canonical scaling strategy for transformer networks and maintain intermediate dimension at twice that of embedding dimension. The impact of these rescalings on next-vertex prediction accuracy is depicted in the two right-most plots of Figure 10. As expected, scale leads to gains in prediction power but these gains quickly saturate. If we rescale our model simultaneously in both directions, we again observe quickly diminishing gains from scale; this result is depicted at the left of Figure 10 and is stated in terms of semantic routing success rate rather than next-edge prediction accuracy. This saturation in gains from scaling up model capacity is one reason to believe that our choice of a simple transformer architecture may not be optimal.

## B   Open Questions

**Longer Trips.**   Most timesteps in real-world routing tasks are uninteresting and, for example, correspond to driving straight on a highway for hundreds of road segments. Typically, this is addressed with hierarchical methods that add "shortcut" edges to summarize common sequences of road segments (Delling et al., 2009). To maintain the simplicity of our proof-of-concept, we only implemented a simple alternative of contracting vertices with edges of the same road type, which only reduced effective road segment length by a factor of 2-3. This posed an issue as, in longer routes, next-vertex pairs contain sparser signals as more of the route is trivial, making training more expensive. We accordingly configured our benchmark to restrict navigation tasks to those can be solved in 600 road segments. The benchmark can be made more challenging by increasing this cap, which will require integrating techniques like strategically adjusting the coarseness of road segments in contraction hierarchies (Delling et al., 2009).

**Alternatives to Next-Vertex Prediction.**   Although we implemented an autoregressive approach that unrolls routes from source to destination, autoregressive strategies need not unroll routes in this order. For example, one can unroll routes from the destination, or in a hierarchical middle-out fashion. As we found that unrolling source to destination resulted in sufficient good performance by real-world measures, we defer consideration of alternative approaches to future work.

**Receptive Field Representation.**   Our model encodes its receptive field inputs in a simple fashion: concatenate all edges into a sequence and pass the sequence through a transformer network so different edges can attend to one another. As our proof-of-concept was not aimed at demonstrating that any particular network architecture is superior, we opted for a basic transformer for simplicity and scalability. We expect that performance improvements can be gained by both enlarging the size of the receptive field and including richer graph metadata (e.g., allowing the model to retrieve the reviews of relevant local businesses during route finding), for which graph representation learning methods such as GNNs and graph transformers (Yun et al., 2019; Dwivedi and Bresson, 2020) are potentially useful thanks to their ability to combine the information from graph connectivity and metadata through learning.

**Efficiency and Scaling.**   The significant attention received by large language models have spurred a number of advancements in implementing autoregressive models, especially transformer networks, to be more efficient and scalable Wang et al. (2020); Zaheer et al. (2020); Dao et al. (2022). As we have shown that autoregressive transformers are indeed viable solutions to semantic routing, these efficiency gains can similarly be realized by semantic routing applications. In particular, the inference efficiency of our model can be greatly improved by adopting architectural modifications like more efficient attention mechanisms (Dao et al., 2022).

## C   Experiment Details

In this section, we review the hyperparameters for each experiment in this paper. In terms of grid world experiments, Table 6a describes the hyperparameters for the point-of-interest density experiment, the beam width experiment and the secondary scoring model ablation experiment; Table 3 describes the hyperparameters for the receptive field size experiment; and Table 5 and Table 4 describe the hyperparameters for the scaling study experiment. The hyperparmeters for the experiment on our benchmark are described in Table 6b. We use "Embedding Dim.", "Intermediate Dim.", and "Attention Heads" to denote the embedding dimension, intermediate dimension, and attention head count of the transformer blocks in the network. Similarly, we use "MLP Layers (Road)" and "Intermediate Dim.

(Road)" to denote the depth and intermediate dimensions of the multi-layer perceptron blocks used to encode road segment features. The "Receptive Field Size" denotes the number of edges provided in the receptive field input. In the "Learning Rate Schedule" key, "Cosine" denotes that a cosine one-cycle learning rate schedule is used. The "Seeds" key refers to the number of random seeds the experiment is performed on. Experiments performed in this paper were conducted using computer resources on Google Cloud. Approximately 10,000 CPU hours were used to generate training data from the benchmark software. The experiments described in Appendix A were performed on a GPU cluster for approximately 1,000 GPU hours. The experiments described in Section 3 were performed on a GPU cluster for approximately 20,000 GPU hours.

| Hyperparameter | Values | | | | | |
|---|---|---|---|---|---|---|
| Datapoints (# Routes) | 1 million | 1 million | 1 million | 1 million | 1 million | 1 million |
| Learning Rate | 0.0001 | 0.0001 | 0.0001 | 0.0001 | 0.0001 | 0.0001 |
| L2 Weight Decay | 0 | 0 | 0 | 0 | 0 | 0 |
| Dropout | - | - | - | - | - | - |
| Batch Size | 1024 | 1024 | 1024 | 1024 | 128 | 128 |
| Iterations | 200,000 | 200,000 | 200,000 | 200,000 | 400,000 | 400,000 |
| Seeds | 5 | 5 | 5 | 5 | 5 | 5 |
| Learning Rate Schedule | Cosine | Cosine | Cosine | Cosine | Cosine | Cosine |
| Receptive Field Size | 8 | 64 | 128 | 256 | 1024 | 3516 |
| Attention Heads | 4 | 4 | 4 | 4 | 4 | 4 |
| Embedding Dim. | 256 | 256 | 256 | 256 | 256 | 256 |
| Intermediate Dim. | 512 | 512 | 512 | 512 | 512 | 512 |
| Transformer Blocks | 6 | 6 | 6 | 6 | 6 | 6 |
| Intermediate Dim. (Road) | 512 | 512 | 512 | 512 | 512 | 512 |
| MLP Layers (Road) | 2 | 2 | 2 | 2 | 2 | 2 |

Table 3: Hyperparameters for the receptive field size experiment on the grid-world dataset.

| Hyperparameter | Values | | | |
|---|---|---|---|---|
| Datapoints (# Routes) | 4 million | 4 million | 4 million | 4 million |
| Learning Rate | 0.0001 | 0.0001 | 0.0001 | 0.0001 |
| L2 Weight Decay | 0 | 0 | 0 | 0 |
| Dropout | - | - | - | - |
| Batch Size | 1024 | 1024 | 1024 | 1024 |
| Iterations | 400,000 | 400,000 | 400,000 | 400,000 |
| Seeds | 5 | 5 | 5 | 5 |
| Learning Rate Schedule | Cosine | Cosine | Cosine | Cosine |
| Receptive Field Size | 256 | 256 | 256 | 256 |
| Attention Heads | 4 | 4 | 4 | 4 |
| Embedding Dim. | 256 | 64 | 1024 | 256 |
| Intermediate Dim. | 512 | 128 | 2048 | 512 |
| Transformer Blocks | 6 | 6 | 6 | 10 |
| Intermediate Dim. (Road) | 512 | 128 | 2048 | 512 |
| MLP Layers (Road) | 2 | 2 | 2 | 2 |

Table 4: Hyperparameters for the scaling study experiment on the grid-world dataset.

| Hyperparameter | | | |
|---|---|---|---|
| Datapoints (# Routes) | 4 million | 4 million | 4 million |
| Learning Rate | 0.0001 | 0.0001 | 0.0001 |
| L2 Weight Decay | 0 | 0 | 0 |
| Dropout | - | - | - |
| Batch Size | 1024 | 1024 | 1024 |
| Iterations | 400,000 | 400,000 | 400,000 |
| Seeds | 5 | 5 | 5 |
| Learning Rate Schedule | Cosine | Cosine | Cosine |
| Receptive Field Size | 256 | 256 | 256 |
| Attention Heads | 4 | 4 | 8 |
| Embedding Dim. | 256 | 64 | 1024 |
| Intermediate Dim. | 512 | 128 | 2048 |
| Transformer Blocks | 6 | 3 | 10 |
| Intermediate Dim. (Road) | 512 | 128 | 2048 |
| MLP Layers (Road) | 2 | 2 | 2 |

Table 5: Hyperparameters for the scaling study experiment on the grid-world dataset (continued).

| Hyperparameter | Values |
|---|---|
| Datapoints (# Routes) | 1 million |
| Learning Rate | 0.0001 |
| L2 Weight Decay | 0 |
| Dropout | - |
| Batch Size | 1024 |
| Iterations | 200,000 |
| Seeds | 5 |
| Learning Rate Schedule | Cosine |
| Receptive Field Size | 256 |
| Attention Heads | 4 |
| Embedding Dim. | 256 |
| Intermediate Dim. | 512 |
| Transformer Blocks | 6 |
| Intermediate Dim. (Road) | 512 |
| MLP Layers (Road) | 2 |

(a) Hyperparameters for grid-world experiments varying point-of-interest density, beam width, and the scoring model.

| Hyperparameter | Values |
|---|---|
| Datapoints (# Routes) | 2 million |
| Learning Rate | 0.00003 |
| L2 Weight Decay | 0 |
| Dropout | - |
| Batch Size | 1024 |
| Iterations | 600,000 |
| Seeds | 3 |
| Learning Rate Schedule | Cosine |
| Receptive Field Size | 512 |
| Attention Heads | 8 |
| Embedding Dim. | 1024 |
| Intermediate Dim. | 2048 |
| Transformer Blocks | 8 |
| Intermediate Dim. (Road) | 2048 |
| MLP Layers (Road) | 2 |

(b) Hyperparameters for the proof-of-concept experiment on our main benchmark, i.e., the results in Table 2.

| Hyperparameter | Values |
|---|---|
| Datapoints | 5,000 |
| Learning Rate | 0.0005 |
| L2 Weight Decay | 0.01 |
| Batch Size | 256 |
| Iterations | 20,000 |
| Intermediate Dim. | 64 |
| Hidden Layers | 1 |
| Activation | GELU |

(c) Hyperparameters for the secondary scoring model in all experiments.

Table 6: Additional hyperparameters.

# D   Example of Predicted Routes

This section provides eight randomly sampled examples of the semantic routing tasks in our benchmark. These are depicted in Figure 11, Figure 12, Figure 13 and Figure 14.

Find me the nearest supermarket to get ingredients for dinner and also suggest a place for a quick lunch. Avoid local roads when possible.

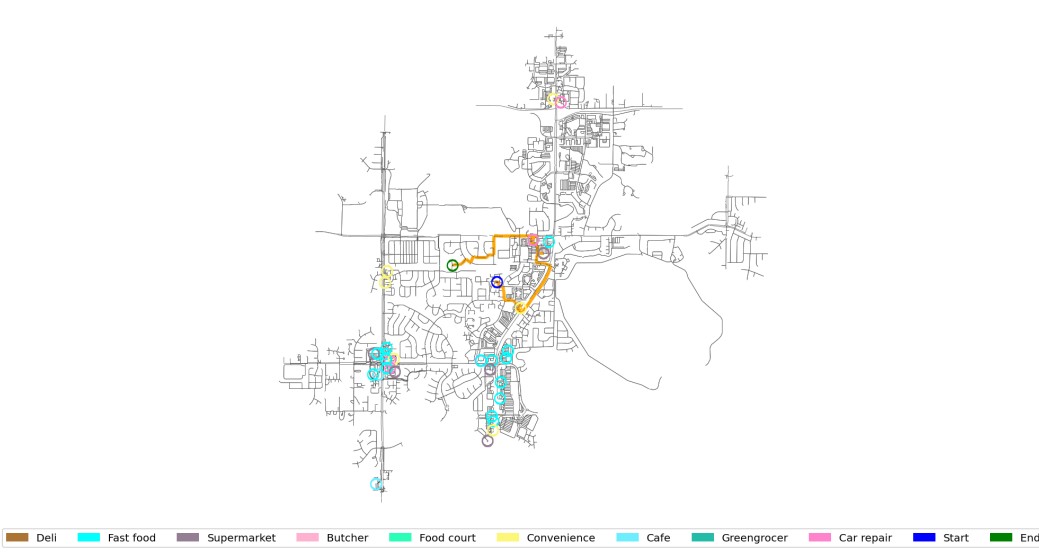

Can you guide me to nearby grocery stores to get dinner ingredients, a gas station to refuel my car, and stores selling energy drinks?

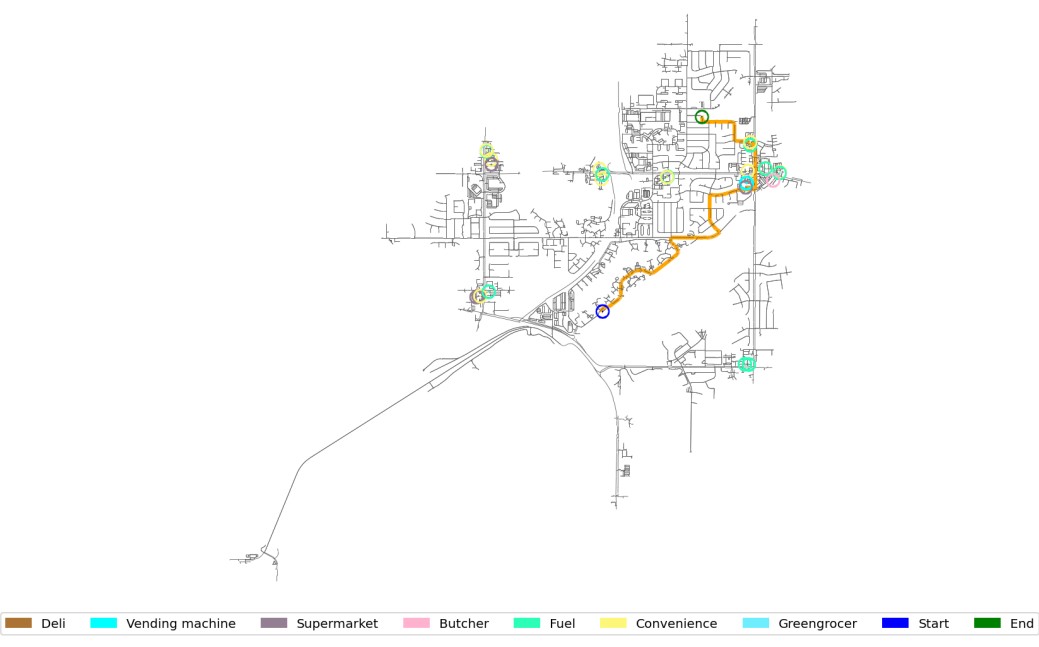

Figure 11: Randomly sampled semantic routing tasks from our benchmark, and their optimal routes (orange).

Suggest a supermarket where I can buy dinner ingredients and also find dessert.

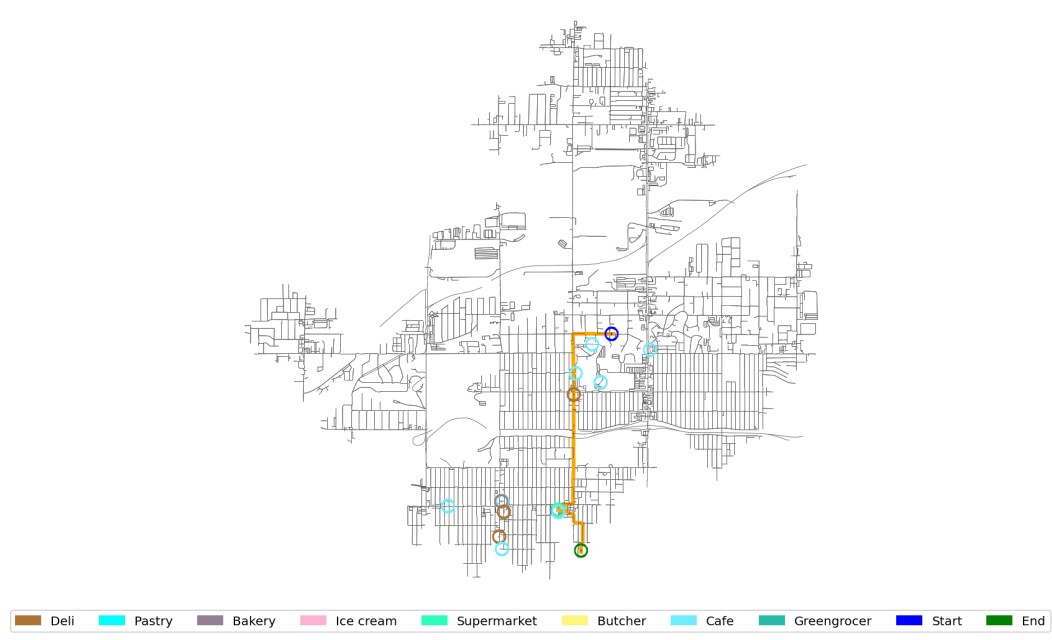

| Deli | Pastry | Bakery | Ice cream | Supermarket | Butcher | Cafe | Greengrocer | Start | End |

Mind finding a nearby pharmacy?

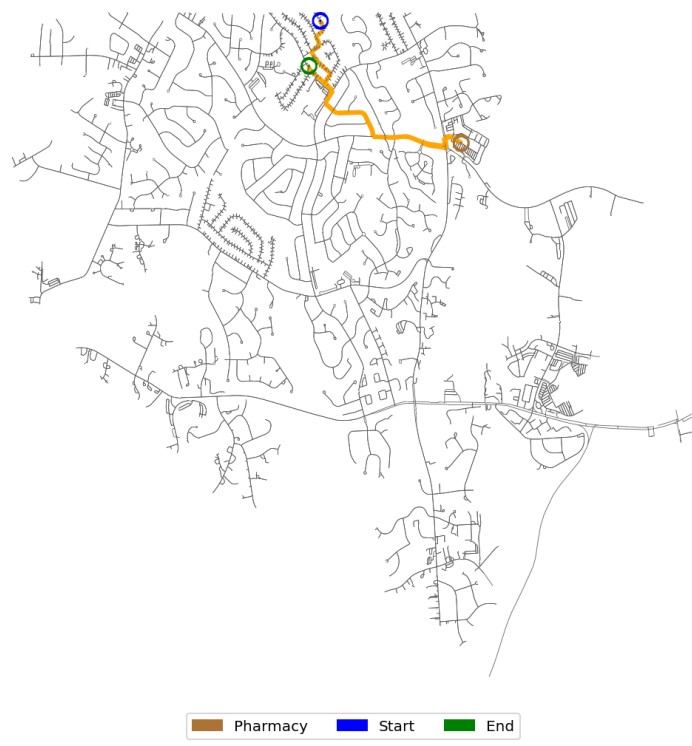

| Pharmacy | Start | End |

Figure 12: Randomly sampled semantic routing tasks from our benchmark, and their optimal routes (orange).

Find the fastest way to a snack store and a fast food place, avoiding highways.

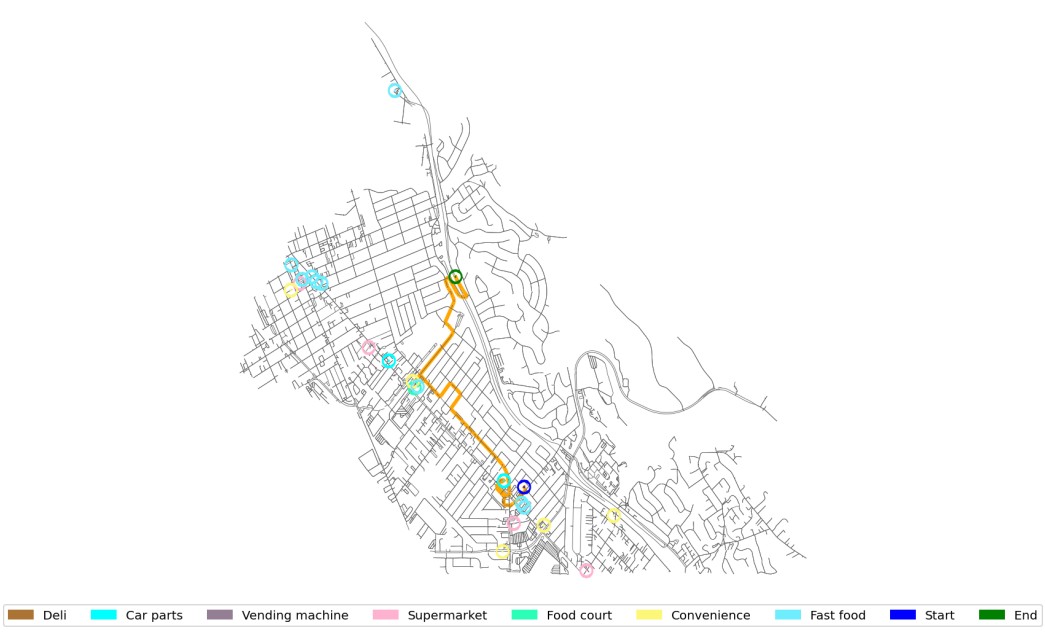

| Deli | Car parts | Vending machine | Supermarket | Food court | Convenience | Fast food | Start | End |

Can you guide me to nearby grocery stores to get dinner ingredients, a gas station to refuel my car, and stores selling energy drinks?

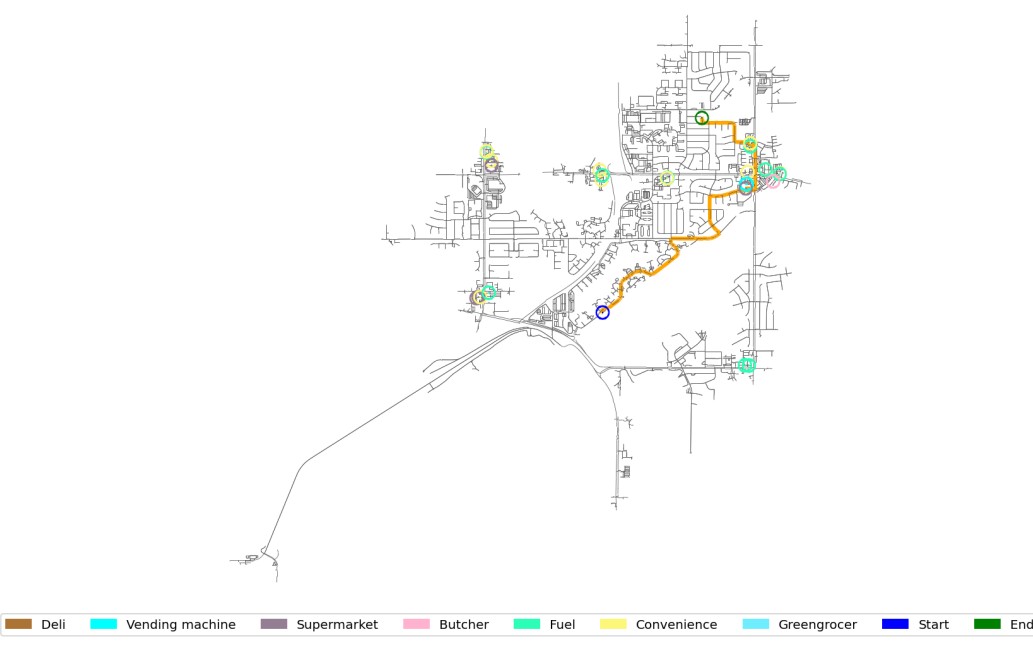

| Deli | Vending machine | Supermarket | Butcher | Fuel | Convenience | Greengrocer | Start | End |

Figure 13: Randomly sampled semantic routing tasks from our benchmark, and their optimal routes (orange).

Recommend a multi-stop adventure around town, with all destinations within a 20-minute drive.

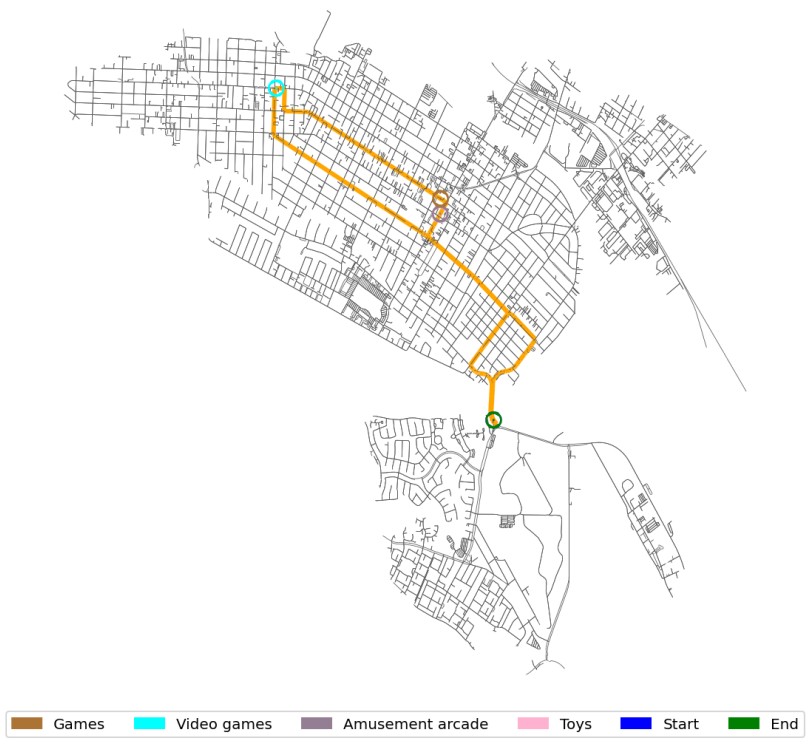

| Games | Video games | Amusement arcade | Toys | Start | End |

I need to stop at shops and buy dinner ingredients, car snacks, and also get gas.

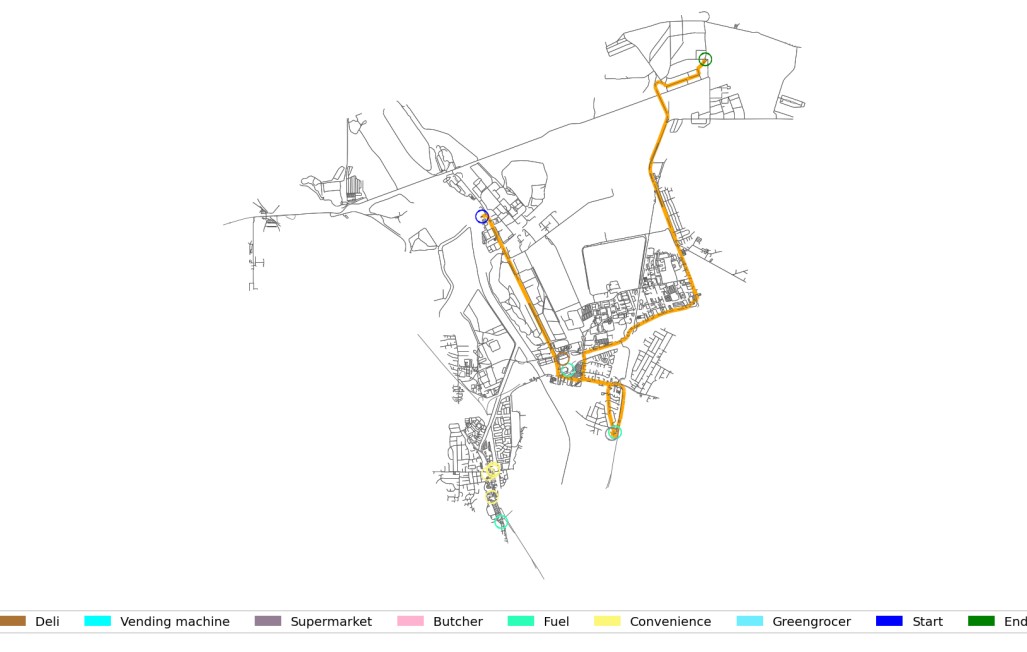

| Deli | Vending machine | Supermarket | Butcher | Fuel | Convenience | Greengrocer | Start | End |

Figure 14: Randomly sampled semantic routing tasks from our benchmark, and their optimal routes (orange).

