# OpenReview forum: "Semantic Routing via Autoregressive Modeling"
_NeurIPS.cc/2024/Conference — NeurIPS 2024 poster_

### Official Review · Reviewer_o9bU · 2024-07-11

**Soundness:** 4
**Presentation:** 3
**Contribution:** 4
**Rating:** 7
**Confidence:** 2

**Summary:**

This paper releases a large-scale routing dataset consisting of diverse multi-objective navigation queries expressed via natural languages on the richly annotated road networks of US cities. They also propose an autoregressive baseline method based on a standard transformer network and show that the baseline is a powerful method on the task.

**Strengths:**

1. The released dataset has a training set along with the test set, making it easy to develop learning-based algorithms on the dataset. The authors also provide an automatic evaluation metric based on human-labeled interpretations of all the queries, which enables automatic evaluation and (possible) reinforcement learning on the dataset
2. The baseline based on the transformer decoder shows promising performance on the benchmark, showing the potential of learning-based methods on the problem.

**Weaknesses:**

1. Originally there are only 1000+ distinct queries, which are augmented into 10,000+ natural language variations by a language model. Multi-objective queries are formed by randomly merging multiple queries and rephrased by the language model. It is not sure whether the augmentation and the rephrasing of the language model are accurate enough to keep the original meanings. This is important since the structured interpretation used for automatic evaluation is based on the original queries.

**Questions:**

1. The introduction section says the dataset has 1 million queries. What is the precise composition of these queries? How many queries are written by humans and how many are augmented by language models? How many queries are multi-objective, i.e. merged from multiple queries?

**Limitations:**

Section B only discusses the limitation of the proposed baseline, not the dataset.

---

> ### Author Rebuttal · Authors · 2024-08-05
>
> We appreciate your feedback and address your remaining questions below.
>
> **Composition of query dataset.**
>
> We appreciate your suggestion for clarifying this point in the benchmark’s construction. We agree that clarifying this point is important, and will incorporate the following answer—along with additional context—into Section 2.2 of our revision:
>
> Details on the composition of the query dataset can be found on L230-L237. The 1,000+ queries were written by manually taxonomizing common road situations (e.g. ran out of gas, passenger spilled drink in backseat), and writing a single representative query for each situation in this taxonomy. A language model was then tasked with finding different ways in which users might write queries in each situation, resulting in 10,000+ variations. The complete dataset was then formed by merging combinations of these queries. As noted on L188-L189, on average, each query in our dataset is the result of merging 1.8 different queries.
>
> **Query dataset augmentation.**
>
> We believe that, in the absence of tens to hundreds of human experts, the use of a language model to augment the queries is a viable (and practical) approach to developing a large-scale weakly supervised dataset. The queries and their structured interpretations are available in the benchmark for manual inspection. Moreover, a random subset of the query dataset was manually spot-checked for correctness. We agree these points are worth clarifying and will add these notes at Line 237.

---

### Official Review · Reviewer_u4su · 2024-07-12

**Soundness:** 3
**Presentation:** 3
**Contribution:** 2
**Rating:** 6
**Confidence:** 3

**Summary:**

The paper proposes an extended semantic routing planning task, which includes user queries expressed in natural language. The author released a large-scale dataset for this task, and proposed a method of using autoregressive models instead of graph-related methods for route planning, which has a significant performance improvement on the benchmark.

**Strengths:**

1. This task is very interesting. Setting waypoints through natural language in route planning has a great demand in real life;
2. A large dataset related to the task;
3. The novel autoregressive method can be regarded as an extension of the multimodal large model.

**Weaknesses:**

1. I have concerns about the evaluation metrics. If the goal is to plan a path that meets user requirements, then the evaluation should consider the length difference between the planned path and the optimal path, as well as the delay in planning methods under this premise, rather than using user satisfaction as the final evaluation metric.
2. It is unclear whether the annotated labels in Figure 2 are provided to the method or only used for the evaluator. If not provided, the baseline method needs a more detailed description of how these natural language parts are utilized.
3. Based on points 1 and 2, is the poor performance of the baseline method due to inadequate adaptation to this new task? For example, reaching points of interest should be considered a prerequisite, and then the shortest path should be searched under this prerequisite, rather than checking if points of interest are included in the shortest path.

**Questions:**

1. How are different input components identified in the autoregressive model?
2. Why is there such a large performance gap between Errands and Locations in Table 2?
3. Beam search has been proven useful in language models. Since only a small receptive field is input into the model in this task, will beam search improve performance?

---

> ### Author Rebuttal · Authors · 2024-08-05
>
> We appreciate your feedback and address your questions below.
>
> **How are different input components identified in the autoregressive model?**
>
> We follow the standard practice of using categorical encodings. That is, in addition to adding a position encoding to each token in the input sequence, we also add a categorical encoding to each token denoting what type of input component the token belongs to.
>
> **Why is there such a large performance gap between Errands and Locations in Table 2?**
>
> Errands correspond to tasks that can be fulfilled by going to multiple possible points-of-interest; for example, water bottles can be found at both gas stations and grocery stores. Hence, simpler solution routes usually exist. However, location tasks correspond to tasks that can only be fulfilled by going to a specific point-of-interest, which makes the tasks more challenging.
>
> **Will beam search improve performance?**
>
> Your intuition is correct and, for the reasons you noted, we do already use beam search in the paper. For example, the reported figures in Table 2 are using beam search. We found that beam search indeed provided a significant boost in performance, which we verified through the ablation experiment plotted in Figure 9.
>
> **Some additional clarifications.**
>
> 1. The “excess penalty” metric generated by our benchmark software does already explicitly factor in travel time. Specifically, “excess penalty” is computed as the excess travel time of the predicted route versus the optimal route, where travel time is scaled up by a factor of 5x along road segments that the user asked to avoid (see Figure 2 caption). We appreciate your highlighting this potential point of confusion and will revise Line 314-318 to be more clear on this point.
>
> 2. The annotated labels in Figure 2 are for the benefit of those reading the paper; the autoregressive model only has access to the natural language form of the queries. The annotated labels are, however, provided to the evaluator, and are also given to the baseline methods, giving the baselines an unfair advantage against our autoregressive model. We will add this note to Line 204 in our revision.
>
> 3. The baseline methods already consider reaching points-of-interest as a prerequisite (L201, L203). The difficulty is that searching under these prerequisites is expensive.

---

> > ### Comment · Reviewer_u4su · 2024-08-12
> >
> > I thank the authors for their clarification and have updated the scores.

---

### Official Review · Reviewer_oN8k · 2024-07-27

**Soundness:** 3
**Presentation:** 3
**Contribution:** 3
**Rating:** 6
**Confidence:** 4

**Summary:**

This paper collects a large-scale benchmark for semantic routing based on road networks of US cities with rich metadata, and 1) shows that the benchmark is challenging for previous methods due to its scale and complex user preferences 2) proposes a learning-based method, where a transformer-based model is trained to autoregressively predict the next plausible road segment based on the query, the current partial route and the neighborhood (receptive field) around the wavefront. The proposed method achieves non-trivial performances, opening up the space for developing stronger learning-based approaches for semantic routing.

**Strengths:**

- The problem of semantic routing is practically useful and technically interesting. The collected benchmark takes a step toward developing systems that can handle problems of realistic difficulty levels.

- The proposed learning-based approach, architecture, and metadata encoding schemes are novel and show impressive transferability to routing problems in unseen cities.

- Further experiments and ablations (though most are in the Appendix) on receptive field size, scoring models, etc. are interesting and could provide insights into future developments in the community.

**Weaknesses:**

- There is a lack of some details on benchmark construction and quantitative statistics. For example, in line 232, how is the "human writing" of queries done? Is it through crowdsourcing via, e.g., Amazon Mechanical Turk? In line 167, "The majority of tasks in our benchmark involve a variant of waypoint routing:..." how much is the "majority"? Similarly for "Another subset of tasks ..." in line 181. These statistics affect the way people think about the examples and the performance here.

- From lines 267-271, it can be seen that the ground-truth optimal routes can actually be found via brute-force search using classical methods in 0.3 CPU hours per route on average - in some sense isn't this suggesting that the problems aren't that difficult after all? Is it possible to do parallelization (this should clearly be possible for the brute-force approach) to speed up the solving process? If the route could be somehow produced in a reasonable time, there seems to be less need for learning-based approaches which have much less transparency and guarantees.

- Some minor ones:
  - The evaluation metric is very tricky as it's mostly based on human heuristics. But I think this is more of an issue of the nature of the task (similar to how it's hard to design reward functions in reinforcement learning). Would be great if it is possible to collect some actual paths that humans prefer to enable learning of a good metric.
  - The writing overall could be improved. For example, the description of the collected benchmarks and the results (baselines, proposed methods) are quite scattered.
  - In the proposed architecture, for the part that encodes the candidate edges, since the model is a decoder transformer, there are unnecessary biases from the orderings of the candidates. May be good to fix this.

**Questions:**

See above.

**Limitations:**

Yes.

---

> ### Author Rebuttal · Authors · 2024-08-05
>
> We thank you for your feedback and address your questions below.
>
> **Details on benchmark construction.**
>
> We appreciate your suggestions on benchmark details that should be elaborated on. We agree these details would be clarifying, and will incorporate the below points—along with additional context—into Section 2.2 of our revision:
>
> The 1,000+ seed queries were written by manually taxonomizing common road situations (e.g. ran out of gas, passenger spilled drink in backseat), and writing a single representative query for each situation in the taxonomy. This taxonomical approach ensures the final dataset provides strong coverage of recorded use-cases. Since this process requires non-trivial domain knowledge, it was performed by the authors and affiliated assistants rather than through a crowdsourcing platform like Mechanical Turk.
>
> To form our benchmark’s training dataset, we collected data on a mixture of tasks where 80% of datapoints correspond to waypoint routing tasks. However, our benchmark returns separate performance statistics for waypoint routing and trip planning tasks (e.g., see the decomposition of Table 2), rather than returning summary statistics for mixtures of tasks.
>
> **Brute-force search complexity.**
>
> The 0.3 CPU hours needed to brute-force a solution is a non-trivial amount of computation for a learning-based method to match, especially since the brute-force algorithm is given unfair access to a structured representation of the user queries that obviates the need to parse the queries in their original natural language form. For this reason, we believe that the benchmark is still a formidable challenge for learning-based methods, as demonstrated in Table 2.
>
> Regarding whether brute-force methods suffice for real-world semantic routing applications, as we elaborate on L54-L59, the motivation for exploring the viability of learning-based semantic routing systems is less about facilitating an otherwise computationally intractable task and more about exploring a scalable approach to supporting diverse natural language queries that require processing large volumes of rich graph metadata. The latter is impractical to scale to real-world application if one's approach is engineering an ensemble of brute-force methods to cover all (or even most) query types.

---

> > ### Comment · Reviewer_oN8k · 2024-08-10
> >
> > Thanks for the response. I have a follow-up question then: the learning-based approach here requires training data including the optimal routes for the query examples, which are obtained by running the brute-force algorithm (lines 267-271). So if the problem is truly intractable, how is the learning-based approach here tractable then? And also why is the "access to a structured representation" unfair since the learning-based approach also depends on it ultimately?

---

> > > ### Author Response · Authors · 2024-08-12
> > >
> > > Thanks for your follow up!
> > >
> > > We would describe the problem as being non-trivial but not intractable. As we noted earlier, our primary motivation for exploring learning-based methods is less about facilitating an otherwise computationally intractable task and more about designing a scalable approach. With that said, we do note that learning-based methods can often efficiently (on average) approximate solutions to problems that are expensive to solve in the worst-case.
> > >
> > > Regarding your second question, our learning-based approach does not depend on having queries that are already parsed into structured data---it needs only a raw natural language user query as input during inference time. This is in contrast to methods, like our baselines, that assume queries already arrive parsed as structured data and would require a human/system to preprocess every user query during inference time; this is why we refer to them as having unfair access.
> > >
> > > You are correct that our benchmark software procedurally generates example routes using brute-force algorithms and a pre-collected and labeled dataset of user queries; and indeed our experiment models were trained on such routes. We provided this feature so that researchers can test new models and approaches on our benchmark without the significant cost of needing to collect human-generated example routes. However, in practice, learning-based systems are trained on already collected private datasets of user queries and their example routes.

---

### Decision · Program_Chairs · 2024-09-25

**Decision:**

Accept (poster)

**Comment:**

This paper studies “semantic route planning” or “semantic routing”, which aims to produce routes in response to rich queries that specify various criteria and preferences, and proposes a learning-based approach by formulating the problem as next step prediction. To facilitate further research, the paper releases a large-scale publicly-licensed benchmark for semantic routing, which consists of real-world multi-objective navigation problems (which are expressed in natural language) on the richly annotated road networks of US cities. Overall the reviewers agree that the problem being addressed in this paper is very interesting, the dataset released will be very useful for future research, and the results of the proposed method are promising. There are some confusions related to details of the benchmark construction and existing approaches being “intractable” or “non-trivial” (which might be due to some places in the paper saying “in addition to being intractable with existing approaches to semantic routing”), so the writing of the paper could be improved. Overall, my recommendation is accept, and that the authors should follow the discussions to refine the paper.